# A Graph Laplacian Eigenvector-based Pre-training Method for Graph Neural Networks

## Abstract

We propose the Laplacian Eigenvector Learning Module (LELM), a novel pre-training module for graph neural networks (GNNs). Traditional message-passing GNNs often struggle to capture global and regional graph structure due to over-smoothing risk as network depth increases. Because the low-frequency eigenvectors of the graph Laplacian matrix encode global information, pre-training GNNs to predict these eigenvectors encourages the network to naturally learn large-scale structural patterns over each graph. Empirically, we show that models pre-trained via our framework outperform baseline models on a variety of graph structure-based tasks. While most existing pre-training methods focus on domain-specific tasks such as feature reconstruction, our self-supervised pre-training framework is structure-based and highly flexible; we show that LELM can be used both as an independent pre-training task and as a plug-in addition to a variety of existing pre-training pipelines.

## 1 Introduction

Graph Neural Networks (GNNs) have become a powerful tool in node and graph representation learning, with successful applications across domains ranging from biomedicine (Cantürk et al., 2023; Yan et al., 2024; Hu et al., 2019; Sun et al., 2022) to social networks (Fan et al., 2019). More recently, graph foundation models (GFMs) are emerging as an exciting field; inspired by the success of large language models (LLMs), researchers are exploring the possibility of creating large graph-based models with emergent capabilities across a wide variety of domains (Liu et al., 2025; Xie et al., 2022; Wang et al., 2025).

A key ingredient in this effort is the creation of self-supervised tasks which can be performed on large unlabelled graph datasets (Liu et al., 2022). A variety of pre-training methods have been proposed, but the majority of such methods are based in contrastive losses and graph reconstruction (Liu et al., 2025; Xie et al., 2022). A few structure-based methods, which precompute labels based on graph topology, have been proposed (Peng et al., 2020a; Hwang et al., 2020). However, this category of pre-training approach, recently termed as *graph property prediction* (Liu et al., 2025), remains unexplored largely due to the limitations of GNNs in capturing global and regional information.

In traditional message passing GNNs, node representations are updated via aggregating neighboring node embeddings. Typically, this involves taking an average or sum of neighboring node embeddings and then passing the new embedding through an MLP. However, for one node to incorporate information from more distant nodes, multiple layers of message passing are required (Alon & Yahav). Increasing the number of layers in a GNN leads to a phenomenon known as oversmoothing: the representations of nodes within a $k$-hop neighborhood become increasingly indistinguishable **?**Oono & Suzuki (2019); Keriven (2022).

We propose LELM, a Laplacian eigenvector-based pre-training module for GNNs and GFMs. Laplacian eigenvectors capture a range of global, regional and local graph structure, making them well-suited as a graph property prediction target. Moreover, LELM utilizes a global MLP prediction head during pre-training that allows the GNN model to learn long-range relationships without requiring excessively deep networks, and augments pre-training data with positional features to overcome expressivity limits of GNNs. LELM is highly flexible: it can be used with any feature types across all graph-based datasets, and can be used both as an independent pre-training method and as a plug-in addition to existing graph pre-training pipelines to improve downstream performance.

Our main contributions are as follows:

1. We introduce LELM as a Laplacian eigenvector-based pre-training module for GNNs.
2. Within LELM, we introduce a global MLP head that enables long-range interaction between vertices within the graph, as well as a set of augmented pre-training features based on the graph diffusion operator.
3. We demonstrate that our pre-training module provides performance improvements over baseline models both as a standalone pre-training task and as an augmentation to existing pre-training pipelines.

## 2 RELATED WORKS

### 2.1 GRAPH PRE-TRAINING METHODS

Towards the goal of improving graph foundation models, a variety of self-supervised graph pre-training tasks have been proposed. According to the taxonomy provided by Liu et al. (2025); Xie et al. (2022), existing graph pre-training methods can be categorized into two broad categories: *contrastive* and *predictive* methods.

Contrastive methods maximize mutual information between pairs of data views using objectives like Jensen-Shannon estimator (Nowozin et al., 2016) or InfoNCE (Oord et al., 2018). Methods can be categorized by the types of views used: graph-node (Sun et al.; Veličković et al.; Peng et al., 2020b), subgraph-node (Hu et al.; Jiao et al., 2020), and subgraph-subgraph Qiu et al. (2020). Some methods also employ graph augmentation to generate two views (You et al., 2020).

Predictive methods, also referred to as generative methods (Liu et al., 2025), self-generate labels and train to predict these labels. A first class of predictive models uses graph reconstruction, whether by using node/edge masking (Xie et al., 2020; Batson & Royer, 2019; Hu et al.) or using autoencoders (Wang et al., 2017; Kipf & Welling, 2016). A second class of predictive methods are *property prediction* methods, which precompute underlying graph properties as labels. Examples include statistical properties such as k-hop connectivity (Peng et al., 2020a) or topological properties like a meta-path (Hwang et al., 2020). Overall, there are a lack of works on property prediction-based methods, with the majority of predictive pre-training methods falling under the former category of graph reconstruction (Liu et al., 2025). Our method, LELM, is the first property prediction method to use the graph Laplacian eigenvectors as a pre-training target.

## 3 BACKGROUND

### 3.1 NOTATION

The unnormalized Laplacian $L$ of a graph $G$ is defined as:

$$L = D - A$$

where $D$ is the diagonal degree matrix and $A$ is the unnormalized adjacency matrix of $G$.

Let $\lambda_1, \lambda_2, \ldots \lambda_k$ denote the $k$ lowest eigenvalues of $L$ in nondecreasing order. Let $\psi_1, \psi_2, \ldots \psi_k$ denote the corresponding eigenvectors, such that we have:

$$L\psi_i = \lambda_i \psi_i$$

Note that for the unnormalized Laplacian, the first eigenvector and eigenvalue are trivial:

$$\psi_1 = \frac{1}{n}\mathbb{1}, \lambda_1 = 0$$

By Courant-Fischer, the eigenvectors of $L$ (and the eigenvectors of any Hermitian matrix) can be equivalently expressed as solutions to the following iterative optimization problem:

$$\psi_k \in \underset{\substack{\|x\|=1 \\ x \perp \psi_1, \ldots, \psi_{k-1}}}{\arg\min} \quad x^\top L x.$$

The term $\frac{x^\top L x}{x^\top x}$ is known as the Rayleigh quotient; because we normalize our predicted eigenvectors, we simply treat this as $x^\top L x$.

## 3.2 GRAPH LAPLACIAN EIGENVECTORS

Our use of the low-frequency graph Laplacian eigenvectors is motivated by their close relationship to structural and positional properties of graphs.

**Provably minimal graph cuts:** The second-lowest eigenvector $\psi_2$, known as the Fiedler vector, can be used to generate a provably "good" cut on a graph; in particular, for some arbitrary threshold $s \in \mathbb{R}$, we can define a Fiedler cut $C$ to be:

$$C = (\{i : \psi_2(i) < s\}, \{i : \psi_2(i) \geq s\})$$

On any bounded-degree $n$-vertex planar graph, the optimal Fiedler cut has ratio $O(\frac{1}{n})$ (Spielman & Teng, 1996).

**Positional encodings:** The low-frequency Laplacian eigenvectors naturally encode a global position on the graph. As a result, Laplacian positional encodings (LapPE) have been used as a standard positional encoding for graph transformer models (Dwivedi & Bresson, 2020; Rampášek et al., 2022). In practice, directly using the Laplacian eigenvectors as positional encodings creates sign and basis ambiguity issues, as $\psi_i$ is an eigenvector of $L \iff -\psi_i$ is an eigenvector of $L$. Approaches to solving this problem include designing an architecture which processes the Laplacian eigenvectors in a sign- and basis-invariant manner (Lim et al., 2022) or defining canonical directions for the eigenvectors (Ma et al., 2023).

**Spectral GNNs:** A variety of methods, known as spectral graph neural networks, use the Laplacian eigendecomposition to learn filters in signal domain (Bo et al., 2023b). Some methods explicitly compute or approximate the $k$ lowest-frequency Laplacian eigenvectors, learning advanced filters on the corresponding eigenvalues (Bruna et al., 2013; Liao et al., 2019; Bo et al., 2023a). Other methods instead learn polynomial filters on the graph (Defferrard et al., 2016; He et al., 2022), circumventing the expensive process of eigendecomposition by learning a $k$-degree polynomial function with respect to $L$, i.e. $p(L) = \theta_0 I + \theta_1 L + \cdots + \theta_k L^k$.

**Spectral clustering:** The Laplacian eigenvectors have also been used for clustering applications. Given a set of data $x_1, \ldots x_n$, Belkin & Niyogi (2001) construct a weighted graph $G$ with $n$ nodes using a heat kernel. Then to generate a $k$-dimensional embedding, Belkin & Niyogi (2001) compute the $k$-lowest eigenvectors $\psi_2, \ldots \psi_{k+1}$ (omitting the trivial eigenvector) of the graph Laplacian and assign data point $x_i$ the embedding $(\psi_2(i), \psi_3(i), \ldots, \psi_{k+1}(i))$. Shaham et al. (2018); Chen et al. (2022) learn this spectral map using a neural network, allowing for a natural extension of this map to new datapoints.

## 4 METHOD

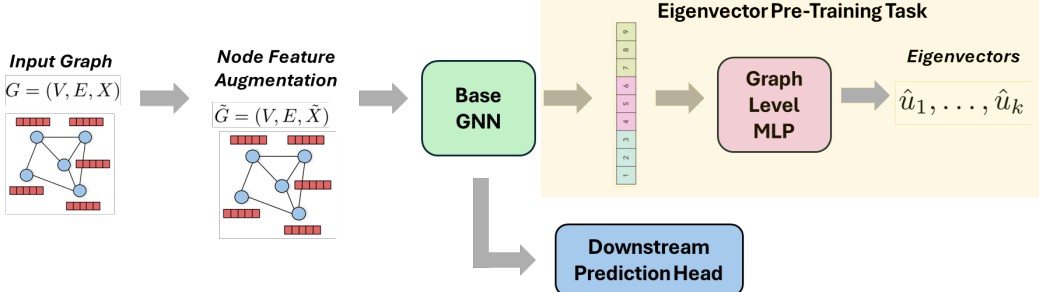

Figure 1: Overview of the LELM pre-training pipeline. Here, "Base GNN" and "Downstream Prediction Head" can be any user-defined model architecture.

## 4.1 OVERVIEW

The LELM pre-training framework consists of three primary components:

- **Node feature augmentation:** We provide initial features based on the diffusion operator: (1) wavelet positional embeddings, and (2) diffused dirac embeddings.
- **Eigenvector prediction:** During pre-training, we task the model to predict the $k$ lowest-frequency eigenvectors of the graph Laplacian.
- **Graph-level MLP:** We pass a graph-level aggregated representation into our prediction MLP head.

## 4.2 Model Architecture

**Base GNN:** The base GNN model takes in a graph with augmented node features and generates learned node representations via neighborhood message passing and update steps. Any GNN architecture may be selected as the base model to fit the needs of the dataset and downstream application.

**Graph-level MLP:** We concatenate the node-wise output of the base GNN model to form a graph-level aggregated representation. We then pass the aggregated vector through an MLP model to produce the low-frequency Laplacian eigenvectors. Concatenating the node embeddings prior to applying the MLP allows the model to learn relationships between distant nodes without risking over-smoothing.

Previous eigenvector-learning methods use a node-wise MLP head, processing each node's eigen-coordinates independently based on their learned hidden embedding (Shaham et al., 2018; Dwivedi et al., 2021; Cantürk et al., 2023).

## 4.3 Node feature augmentation

To provide the model with additional structural information, and to overcome well-known expressivity limits of GNNs (Morris et al., 2019; Maron et al., 2019; Xu et al., 2018), we augment node features with structure-based embeddings. We propose two kinds of embeddings: **(1) wavelet positional embeddings**, which encode relative positional information between nodes, and **(2) diffused dirac embeddings**, which encode local connectivity structures around each node. Both embeddings use the random walk matrix, and capture local aggregate information on each node. The diffusion operator $P$ of a graph $G$ is defined as:

$$P = D^{-1}A$$

Each entry $P_{ij}$ represents the probability of starting a random walk at node $i$ and ending at node $j$ after one step. One can also take powers of the diffusion operator, $P^t$. Each entry of the powered matrix, $P_{ij}^t$, represents the probability of starting a random walk at node $i$ and ending at node $j$ after $t$ steps

The $j^{\text{th}}$ wavelet operator $\Psi_j$ of a graph $G$ is defined as:

$$\Psi_j = P^{2^{j-1}} - P^{2^j}$$

$$\Psi_0 = I - P$$

A wavelet bank, $\mathcal{W}_J$ is a collection of wavelet operators such that:

$$\mathcal{W}_J = \{\Psi_j\}_{0 \leq j \leq J} \cup P^{2^J}$$

**Wavelet positional embeddings** encode information about the relative position of each node within the graph. We randomly select two nodes from each graph, $i$ and $j$, and start dirac signals $\delta_i, \delta_j$. We then apply these signals to each wavelet, $\Psi_k$, in our wavelet bank. The wavelet positional embedding for node $m$ is the $m^{\text{th}}$ row of the resulting matrix.

$$w_{m,k} = \Psi_k(m, \cdot) \begin{bmatrix} | & | \\ \delta_i & \delta_j \\ | & | \end{bmatrix}$$

$$w_m = [w_{m,1} \quad \ldots \quad w_{m,J}]$$

**Diffused dirac embeddings** encode information about the connectedness of each node and its neighbors. For each node, $m$, we apply the $m^{\text{th}}$ row of the diffusion matrix $P$ to each wavelet $\Psi_k$ in our

wavelet bank. As above, the diffused dirac embedding for node $m$ is the $m^{\text{th}}$ row of the resulting matrix.

$$d_{m,k} = \Psi_k(m, \cdot) \, P(m, \cdot)^\top$$

$$d_m = [d_{m,1} \quad \ldots \quad d_{m,J}]$$

These node embeddings are unique up to co-spectrality of the graph Laplacian. The proof is provided in A.5.

## 4.4 LOSS FUNCTION

Let $\hat{U}$ denote a matrix of $k$ column vectors $\hat{u}_i$, where each $\hat{u}_i$ denotes the $i$th predicted eigenvector. Let $\Lambda_k$ denote a diagonal matrix containing eigenvalues $\lambda_1, \ldots, \lambda_k$.

We minimize a weighted sum of two loss functions: **(1) eigenvector loss** and **(2) energy loss**. Both loss functions respect necessary sign and basis invariances of Laplacian eigenvectors; full proofs can be found in A.4.

To ensure the model does not output $k$ copies of the trivial eigenvector, we impose orthogonality on the final outputs of the model via QR decomposition, as proposed by Shaham et al. (2018).

**Energy loss**, used by Shaham et al. (2018); Dwivedi et al. (2021); Ma & Zhan (2023), aims to minimize the sum of Rayleigh quotients:

$$\mathcal{L}_{energy} = \frac{1}{k} \, \text{Tr}(\hat{U}^\top L \hat{U})$$

This loss function is motivated by the iterative optimization problem following from Courant-Fischer (3.1). However, minimizing this loss function only minimizes the *sum* of the first $k$ Rayleigh quotients, meaning the minimizer (subject to orthogonality) is any set of vectors spanning same subspace spanned by the $k$ lowest frequency eigenvectors. For applications in clustering, this is reasonable, as the exact basis in which embeddings are expressed is often irrelevant; however, to require the model to truly predict the $k$-lowest eigenvectors, we must include a more explicit penalty, such as **eigenvector loss**.

**Eigenvector loss**, used by Mishne et al. (2019), measures the difference between each $L\hat{u}_i$ and $\lambda_i \hat{u}_i$:

$$\mathcal{L}_{eigvec} = \frac{1}{k} \|(L\hat{U} - \hat{U}\Lambda_k)\|$$

Eigenvector loss enforces both the correct basis and a strict ordering (up to eigenvalue multiplicity) on the predicted eigenvectors. Our final loss function is then:

$$\mathcal{L} = \alpha \cdot \mathcal{L}_{energy} + \beta \cdot \mathcal{L}_{eigvec}$$

## 4.5 PRE-TRAINING ALGORITHM

---
**Algorithm 1** Eigenvector Prediction

---
**Require:** Graph $G = (V, E)$; augmented node features $\tilde{X} = \{\tilde{x}_j\}$; Base GNN
**Ensure:** Output Pre-trained GNN model, $k$ lowest-frequency eigenvectors
 1: **for** $i <$ Pre-Training Epochs **do**
 2: $\quad \vec{z}_0, \ldots, \vec{z}_n \leftarrow \text{BASEGNN}(G, \tilde{X})$
 3: $\quad \vec{Z} \leftarrow [\vec{z}_1, \ldots, \vec{z}_n] \in \mathbb{R}^{nd}$
 4: $\quad \tilde{U} \leftarrow \text{MLP}(\vec{Z})$
 5: $\quad \hat{U} = \text{QR}(\tilde{U})$
 6: $\quad Loss = \alpha \cdot \text{ENERGYLOSS}(\hat{U}) + \beta \cdot \text{EIGVECLOSS}(\hat{U})$
 7: $\quad$ Back-propagate Loss, update model weights
 8: **end for**
 9: **return** BASEGNN

---

## 5 EXPERIMENTAL RESULTS

To evaluate the effectiveness of our framework, we conduct experiments across multiple graph learning use-cases. First, we apply our pre-training framework directly to several GNN models and assess its impact on downstream performance. Second, we integrate our pre-training method with existing pre-training frameworks and examine how our approach can complement established methods. Finally, we pre-train a positional and structural encoder for Graph Transformer networks. Collectively, these approaches provide a comprehensive assessment of the framework's effectiveness across different graph learning scenarios.

### 5.1 PRE-TRAINING GRAPH NEURAL NETWORKS

We pre-train a standard Graph Isomorphism Network (GIN) (Xu et al., 2019) and GPS, a graph transformer using LELM. Once the model has been pre-trained, we replace the graph-level MLP head with a downstream prediction MLP and fine-tune model weights. We evaluate our pre-training framework on three molecular datasets ZINC, ZINC-12k (Sterling & Irwin, 2015) and QM9 (Ramakrishnan et al., 2014). For each of these models, we compare LELM against the same GNN model without pre-training. For each of these models, pre-training improves performance for all but one of the downstream targets. We record results of our experiments in Table 1. In addition, for the GIN model we compare LELM to various structure-based pre-training targets including node degree, local clustering coefficient, random walk structural encodings, cycle counting, and Laplacian eigenvalues. Results are recorded in Table 2.

Table 1: Test MAE (↓) performance comparison on ZINC (single metric) and QM9 (first seven target properties).

| Model | ZINC full Penalized $\log p$ | ZINC subset Penalized $\log p$ | QM9 $\mu$ | $\alpha$ | $\varepsilon_{\text{HOMO}}$ | $\varepsilon_{\text{LUMO}}$ | $\Delta_\varepsilon$ | $R^2$ | ZPVE |
|---|---|---|---|---|---|---|---|---|---|
| Pre-Trained GIN | 0.130 | 0.353 | 0.484 | **0.489** | **0.00353** | **0.00371** | 0.00513 | **28.103** | **0.000477** |
| GIN (baseline) | 0.228 | 0.438 | 0.472 | 1.132 | 0.00386 | 0.00399 | 0.00562 | 50.909 | 0.002400 |
| Pre-Trained GPS | **0.104** | **0.210** | 0.5021 | 0.5922 | 0.0037 | 0.0040 | **0.0051** | 33.606 | 0.00178 |
| GPS (baseline) | 0.150 | 0.358 | **0.413** | 0.718 | 0.00434 | 0.00442 | 0.00592 | 80.503 | 0.00111 |

Table 2: Test MAE (↓) performance comparison on ZINC with alternative structural targets.

| Alternative targets | ZINC full | ZINC subset |
|---|---|---|
| LELM | **0.130** | **0.353** |
| Node degree | 0.238 | 0.471 |
| Local clustering coefficient | 1.493 | 1.551 |
| RWSE (Dwivedi) | 1.493 | 1.551 |
| Cycle counting | 0.285 | 0.420 |
| Lap Eigenvalues | 0.250 | 0.520 |

### 5.2 ENHANCING AN EXISTING GRAPH NEURAL NETWORK PRE-TRAINING METHOD

We augment the existing molecular pre-training methods proposed by Hu et al. (2019) with eigenvector-learning. In particular, Hu et al. (2019) propose node-level pre-training tasks (context prediction and masking) on ZINC15 (Sterling & Irwin, 2015), followed by a graph-level supervised pre-training task on ChEMBL (Mayr et al., 2018; Gaulton et al., 2012). We augment the graph-level supervised pre-training step by adding an additional MLP head to predict eigenvectors, and we evaluate on five downstream datasets based on work by Sun et al. (2022).

Detailed results are shown in Table 3. Eigenvector-learning consistently improves performance for the masking pre-training pipeline, but achieves mixed results on the context prediction pipeline. Notably, performance for the masking pipeline was increased for all five datasets when performing eigenvector pre-training with the graph-level MLP head.

Table 3: Test ROC-AUC (%, ↑) performance on 5 molecular prediction tasks when **augmenting an existing pre-training method** on a GIN base model. *Sup.* refers to the original supervised pre-training as implemented by Hu et al. (2019), and *Sup.+* refers to supervised training with LELM. Results for *no pre-training* are taken directly from Sun et al. (2022). All methods are tuned over seven learning rates and averaged over three seeds.

| Dataset | | BACE | BBBP | Tox21 | ToxCast | SIDER |
|---|---|---|---|---|---|---|
| Pretrain method | MLP Head | | | | | |
| ContextPred, Sup.+ | Graph-level | $79.62 \pm 3.63$ | $\mathbf{70.76 \pm 1.64}$ | $77.94 \pm 0.11$ | $\mathbf{66.13 \pm 0.34}$ | $60.05 \pm 0.99$ |
| ContextPred, Sup.+ | Node-wise | $75.87 \pm 3.11$ | $68.74 \pm 1.07$ | $78.86 \pm 0.06$ | $63.78 \pm 0.32$ | $59.83 \pm 0.53$ |
| ContextPred, Sup. | - | $\mathbf{84.98 \pm 1.28}$ | $68.25 \pm 0.48$ | $77.44 \pm 0.19$ | $64.01 \pm 0.81$ | $\mathbf{62.87 \pm 0.89}$ |
| Masking, Sup.+ | Graph-level | $80.71 \pm 3.84$ | $68.33 \pm 0.89$ | $79.09 \pm 0.25$ | $65.96 \pm 0.20$ | $62.41 \pm 1.77$ |
| Masking, Sup.+ | Node-wise | $81.02 \pm 1.67$ | $69.94 \pm 1.76$ | $\mathbf{79.33 \pm 0.41}$ | $65.14 \pm 0.44$ | $59.38 \pm 1.11$ |
| Masking, Sup. | - | $75.42 \pm 2.64$ | $67.36 \pm 4.60$ | $78.33 \pm 0.24$ | $64.88 \pm 0.82$ | $61.6 \pm 1.78$ |
| No pre-training | - | $75.77 \pm 4.29$ | $69.62 \pm 1.05$ | $75.52 \pm 0.67$ | $63.67 \pm 0.32$ | $59.07 \pm 1.13$ |

## 5.3 PRE-TRAINING STRUCTURAL ENCODER

Table 4: Test MAE (↓) performance on ZINC (12k subset) dataset when **augmenting a graph structural encoder**. All results using no structural encoder or the base GPSE are taken directly from Cantürk et al. (2023). *GPSE+* refers to GPSE with LELM. Our experimental results are averaged over three seeds.

| Base model | Structural encoder | MLP Head | MAE |
|---|---|---|---|
| GPS | GPSE+ | Graph-level | $\mathbf{0.0629 \pm 0.0016}$ |
| | GPSE+ | Node-wise | $0.0663 \pm 0.0024$ |
| | GPSE | - | $0.0648 \pm 0.0030$ |
| | None | - | $0.118 \pm 0.005$ |
| GIN | GPSE+ | Graph-level | $0.1231 \pm 0.0026$ |
| | GPSE+ | Node-wise | $0.1299 \pm 0.0010$ |
| | GPSE | - | $0.124 \pm 0.002$ |
| | None | - | $0.285 \pm 0.004$ |

We modify Graph Positional and Structural Encoder (GPSE) (Cantürk et al., 2023). While GPSE already incorporates eigenvector-learning by including Laplacian positional encodings (LapPE) as a prediction target, GPSE uses MAE and cosine similarity loss on the absolute value of each eigenvector and a node-wise prediction head for every node property. We replace GPSE's eigenvector-learning component with our own, using a separate MLP head and eigenvector loss. We keep all other GPSE model settings the same.

Following Cantürk et al. (2023), we pre-train our modified GPSE model on MolPCBA (Hu et al., 2020). We evaluate the effectiveness of these encodings by training both the Transformer model GPS (Rampášek et al., 2022) and a standard GIN, augmented with these encodings, on the downstream molecule property prediction task for the ZINC 12k (Sterling & Irwin, 2015) subset.

We report results on the effectiveness of these new learned encodings in Table 4. Modified eigenvector-learning with a graph-level MLP improves performance on downstream performance for both the GIN and GPS models, with our graph-level GPS+GPSE+ configuration achieving SOTA performance over all model and encoding configurations tested by Cantürk et al. (2023). We also demonstrate that our choice of loss function is crucial for eigenvector-learning in A.9.

## 6 LIMITATIONS AND FUTURE WORK

There are several promising future directions toward improving the Laplacian eigenvector pre-training framework. We have demonstrated the effectiveness of the framework for pre-training and fine-tuning a GNN on the same dataset, or on domain-related datasets. However, we are yet to explore the effectiveness of eigenvector pre-training as a *transfer learning* framework.

Further, the practical implementation of the graph-level MLP requires adding padding to the concatenated node embeddings to accommodate for graphs of differing sizes. This creates an additional challenge for the MLP to learn meaningful relationships between the individual node embeddings within the graph-level vector. One could explore other ways of creating a rich graph-level representation while avoiding this challenge.

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

# A APPENDIX

## A.1 LLM USAGE

We used ChatGPT-5 to give our final paper a readthrough and check for blatant typos and errors. Here is the prompt used, which accompanied our attached (anonymous) paper draft:

> Check for any blatant typos or mistakes, and point to exact page numbers or line numbers. Do not make suggestions on any other aspect of the paper.

## A.2 FULL EIGENVECTOR PRE-TRAINING PIPELINE

We provide a broad algorithmic outline of eigenvector pre-training process in Algorithm 2.

---

**Algorithm 2** Structure-Informed Graph Pre-training Framework

---

**Input:** Graph $G = (V, E)$; node features $X = \{x_j\}$; training labels $Y$; untrained Base GNN; untrained Downstream Prediction Head
**Output:** Trained Base GNN and Downstream Prediction Head
 1: $\tilde{X} \leftarrow$ AUGMENTFEATURES$(G, X)$
 2: BASEGNN $\leftarrow$ EIGVECPRETRAIN$(G, \tilde{X},$ BASEGNN$)$
 3: **for** $i <$ Fine-tuning Epochs **do**
 4:     $\vec{z}_0, \ldots, \vec{z}_n \leftarrow$ BASEGNN$(G, \tilde{X})$
 5:     $\vec{Z} \leftarrow [\vec{z}_1, \ldots, \vec{z}_n]$
 6:     $\hat{Y} \leftarrow$ DOWNSTREAMHEAD$(\vec{Z})$
 7:     $Loss =$ LOSSCRITERION$(\hat{Y}, Y)$
 8:     Backpropagate Loss, update model weights
 9: **end for**
10: **return** BASEGNN, DOWNSTREAMHEAD

---

## A.3  LOSS FUNCTION

Eigenvector loss, per-vector form:

$$\mathcal{L}_{eigvec} = \frac{1}{k}\sum_{i=1}^{k}\|L\hat{u}_i - \lambda_i\hat{u}_i\|$$

Eigenvector loss, matrix form:

$$\mathcal{L}_{eigvec} = \frac{1}{k}\|(L\hat{U} - \hat{U}\Lambda_k)\|$$

Energy loss, per-vector form:

$$\mathcal{L}_{energy} = \frac{1}{k}\sum_{i=1}^{k}\hat{u}_i^{\top}L\hat{u}_i$$

Energy loss, matrix form:

$$\mathcal{L}_{energy} = \frac{1}{k}\operatorname{Tr}(\hat{U}^{\top}L\hat{U})$$

Energy loss is order-invariant and rotation invariant (see A.4); for applications in clustering, this is reasonable. However, we would like the model to learn the eigenvectors in their specific order, so we also define **absolute energy loss**, matching the Rayleigh quotient with the ground-truth eigenvalue:

$$\mathcal{L}_{energy\_abs} = \frac{1}{k}\sum_{i=1}^{k}|\hat{u}_i^{\top}L\hat{u}_i - \lambda_i|$$

This can be written as, in matrix form:

$$\mathcal{L}_{energy\_abs} = \frac{1}{k}\operatorname{Tr}|\hat{U}^{\top}L\hat{U} - \Lambda_k|$$

In practice, we do not show any results using absolute energy loss, and instead linearly combine energy loss with eigenvector loss to avoid order and rotation invariance. However, absolute energy loss remains an interesting avenue to explore.

### A.3.1  ORTHOGONALITY

To ensure the model does not output $k$ copies of the trivial eigenvector, we must give the model orthogonality constraints on the output vectors. There are again two reasonable choices here: **(1) forced orthogonality** and **(2) orthogonality loss**.

**Forced orthogonality**, used in Shaham et al. (2018), imposes orthogonality on the final outputs of the model via QR decomposition. In other words, if $\hat{U}'$ is the initial output to the model, $Q$ is an $n \times k$ matrix with orthonormal columns, and $R$ is a $k \times k$ upper triangular matrix, then we achieve the final output $\hat{U}$ as such:

$$QR = \hat{U}'$$
$$\hat{U} = Q$$

**Orthogonality loss**, used in Dwivedi et al. (2021); Ma & Zhan (2023); Mishne et al. (2019) imposes a softer constraint, encouraging orthogonality by penalizing the model for producing pairwise similar vectors. This can be written as:

$$\mathcal{L}_{ortho} = \frac{1}{k}\|\hat{U}^{\top}\hat{U} - I\|$$

Based on preliminary testing, we found that forced orthogonality improved performance on the eigenvector-learning, and thus use forced orthogonality in all of our experiments.

## A.4 ENERGY AND EIGENVECTOR LOSSES ARE SIGN AND BASIS INVARIANT

### A.4.1 DEFINITION OF BASIS INVARIANCE

Consider any eigenspace spanned by ground truth eigenvectors $[\psi_j, \psi_{j+1}, \dots \psi_{j+k-1}] = V$. Also recall that, by Spectral Theorem, we can decompose any vector $u$ into a linear combination of all eigenvectors:

$$u = \sum_{i=1}^{n} c_i \psi_i$$

Then a loss function is basis invariant if any rotation of the projected component $VV^\top u$ does not change the loss incurred by $u$. In other words, $u$ gets to arbitrarily "choose" with what basis it wants to express its $VV^\top u$ component. Sign invariance is a special case of basis invariance, where changing sign is equivalent to rotating over a one-dimensional subspace (note that this is slightly stronger than the most apparent form of sign invariance, where we would say $\mathcal{L}(u) = \mathcal{L}(-u)$; instead, we can flip any *component* $c_i \psi_i$ of $u$ when decomposed in terms of eigenvectors).

**Definition 1** (Basis invariance). *Consider an eigenspace spanned by ground truth eigenvectors* $[\psi_j, \psi_{j+1}, \dots \psi_{j+k-1}] = \Psi \in \mathbb{R}^{n \times k}$. *Consider an eigenspace rotation $R_\Psi$ defined as such:*

$$R_\Psi = \Psi A \Psi^\top + (I_n - \Psi \Psi^\top), A \in SO(k)$$

*A loss function $\mathcal{L}(u)$ is basis invariant if, for all such $\Psi, R_\Psi, u \in \mathbb{R}^n$, we have:*

$$L(u) = L(R_\Psi u)$$

### A.4.2 PROOFS

**Lemma 1** (Energy loss is basis invariant). *For any $R_\Psi$ and a single eigenvector prediction $u \in \mathbb{R}^n$, we have:*

$$u^\top L u = (R_\Psi u)^\top L (R_\Psi u)$$

*Proof.* First note that $R_\Psi$ is orthogonal; the set of all $R_\Psi$ describes a subset of $SO(k)$ where only the $k$ basis vectors in $\Psi$ are rotated. Thus, we have $R_\Psi^\top R_\Psi = I$.

In addition, because $\Psi$ is an eigenspace, all columns are eigenvectors with a shared eigenvalue $\lambda$. Then we have:

$$R_\Psi L = \Psi A \Psi^\top L + L - \Psi \Psi^\top L = \lambda \Psi A \Psi^\top + L - \lambda \Psi \Psi^\top = L \Psi A \Psi^\top + L - L \Psi \Psi^\top = L R_\Psi$$

Then we have:

$$R_\Psi^\top L R_\Psi = R_\Psi^\top R_\Psi L = L$$

Thus, for any $u$, we have:

$$u^\top L u = u^\top R_\Psi^\top L R_\Psi u = (R_\Psi u)^\top L (R_\Psi u)$$

$\square$

**Lemma 2** (Eigenvector loss is basis invariant). *For any $R_\Psi$ and a single eigenvector prediction $u \in \mathbb{R}^n$ and ground truth eigenvalue $\lambda$, we have:*

$$\|L u - \lambda u\| = \|L(R_\Psi u) - \lambda(R_\Psi u)\|$$

*Proof.* We know, from our proof above in Lemma 1, that $R_\Psi L = L R_\Psi$. Because $R_\Psi \in SO(k)$, we have $\|R_\Psi x\| = \|x\|$ for any $x \in \mathbb{R}^n$. Then we have:

$$\|L u - \lambda u\| = \|R_\Psi(L u - \lambda u)\|$$
$$\|L u - \lambda u\| = \|L(R_\Psi u) - \lambda(R_\Psi u)\|$$

$\square$

We have an even stronger statement of invariance for energy loss: **given a predicted set of $k$ orthogonal vectors, rotating the vectors within the same subspace does not impact loss.** In other words, a model trained on energy loss only needs to predict the correct *subspace* of $k$ eigenvectors. This is clearly not true of eigenvector loss. Depending on the application, this kind of invariance can be good or bad.

**Lemma 3** (Energy loss is rotation invariant). *Let $L$ be a Laplacian matrix and $V \subseteq \mathbb{R}^n$ be some $k$-dimensional subspace. Suppose $U = [u_1, u_2, \ldots, u_k], W = [w_1, w_2, \ldots, w_k] \in \mathbb{R}^{n \times k}$ are both orthonormal bases for $V$. Then we have:*

$$\frac{1}{k} \operatorname{Tr}(U^\top L U) = \frac{1}{k} \operatorname{Tr}(W^\top L W)$$

*Proof.* Note that $UU^\top = WW^\top$, as they are both orthogonal projectors for the same subspace. Then we have, by the cyclic property of trace:

$$\frac{1}{k} \operatorname{Tr}(U^\top L U) = \frac{1}{k} \operatorname{Tr}(UU^\top L) = \frac{1}{k} \operatorname{Tr}(WW^\top L) = \frac{1}{k} \operatorname{Tr}(W^\top L W)$$

$\square$

### A.5 NODE FEATURE AUGMENTATION

**Lemma 4** (Uniqueness up to co-spectrality). *Let $G_1, G_2$ be graphs of size $n$ with Laplacian matrices $L_1, L_2$ respectively. Let $d_m^1, d_m^2$ represent the diffused dirac embeddings for each node in $G_1, G_2$. Then if $L_1$ and $L_2$ have different eigenvalues, $\{d_m^1 : m \le n\} \ne \{d_m^2 : m \le n\}$*

*Proof.* Consider the random-walk Laplacian of a graph: $L_{rw} := I - D^{-1}A = I - P$. Moreover, note that $L_{rw} = D^{-1}L$. Observe that

$$\begin{aligned} L_{rw}Dv &= D^{-1}LDv \\ &= D^{-1}U\Lambda U^\top Dv \\ &= Bv \text{ for some diagonalizable matrix } B \text{ with eigenvalues } \lambda_i, \ldots, \lambda_n \end{aligned}$$

Where $U = [\psi_1 \quad \ldots \quad \psi_n]$, with $\psi_i$ orthonormal eigenvectors of $L$ and $\Lambda$ is the diagonal matrix of eigenvalues $\lambda_1, \ldots, \lambda_n$ of $L$. Any change to the eigenspectrum of $L$, clearly results in a change to $L_{rw}$, and therefore $P$. Since $\Psi_0 = I - P$, any two graphs with distinct Laplacian eigenspectra will have distinct diffused dirac node embeddings. $\square$

### A.6 DETAILED EXPERIMENTAL SETTINGS

A complete overview of model hyperparameters and settings can be found in Table 5. Heuristically, the Graph-level MLP head hidden dimension is chosen to be the max # nodes multiplied by the hidden dimension size of the base GNN. We do NOT omit the trivial eigenvector when counting number of eigenvectors predicted.

### A.7 ALTERNATIVE STRUCTURAL PRE-TRAINING TARGETS

Here, we formally define and provide details for the alternative pre-training targets used in section 5.1.

- **Node degree:** A node-level label representing the degree of each node
- **Local clustering coefficient:** A node-level label computing the local clustering coefficient of each node. For a fixed node $u$, the coefficient $C$ is given by:

$$C = \frac{2|\{e_{jk} : v_j, v_k \in N_u, e_{jk} \in E\}|}{|N_u|(|N_u| - 1)}.$$

- **RWSE:** A node-level label computing self-walk probabilities at varying step counts for the diffusion operator (Dwivedi et al.). In our experiments, we use step counts from the interval $[2, 22]$.
- **Cycle counting:** A graph-level label computing cycle counts of varying lengths. In our experiments, we count cycles up to length 9.

- **Lap Eigenvalues:** A graph-level label computing the $k$-lowest Laplacian eigenvalues $\lambda_1, \ldots, \lambda_k$. We use the same $k = 6$ as we do with LELM.

For all alternative structural pre-training tasks, we use the same hyperparameters for GIN as displayed in 5, with no initial features and using a standard MAE loss instead of eigenvector + energy loss. We train on the full ZINC dataset. All structural pre-training targets are normalized to have mean 0 and standard deviation 1 across the entire dataset.

Table 5: A comprehensive list of all model hyperparameters used during the eigenvector pre-training step. All hyperparameters highlighted in gray are specific to eigenvector-learning, while other listed configs reflect general GNN settings (and are set to match default values in each respective baseline work).

| Method | GIN (5.1) | GPS (5.1) | GIN pre-training (5.2) | GPSE (5.3) |
|---|---|---|---|---|
| Pre-training dataset | ZINC-subset (12k), ZINC (250k), QM9 (134k) | ZINC-subset (12k), ZINC (250k), QM9 (134k) | ZINC15 (2M), ChEMBL (456K) | MolPCBA (324K) |
| Base architecture | GIN | Transformer/GIN | GIN | MPNN |
| # params | 33543 | 157680 | 2252210 | 22075899 |
| # layers of per-node feature update | 3 | 3 | 2 | 1 |
| # layers of message passing | 4 | 4 | 5 | 20 |
| Hidden dim | 60 | 60 | 300 | 512 |
| Activation fn | ReLU | ReLU | ReLU | ReLU |
| Dropout | 0.1 | 0.1 | 0.2 | 0.2 |
| Batch size | 128 | 128 | 32 | 1024 |
| Learning rate | 0.001 | 0.001 | 0.001 | 0.005 |
| Optimizer | Adam | Adam | Adam | AdamW |
| Scheduler | ReduceLROnPlateau | ReduceLROnPlateau | None | CosineWithWarmup |
|  | patience=5, factor=0.9 | patience=20, factor=0.5 | – | – |
| Pre-Training Epochs | 200 | 100 | 100 | 120 |
| Fine Tuning Epochs | 500 | 150 | 100 | - |
| **Laplacian norm type** | Unnormalized | Unnormalized | Unnormalized | Symmetric |
| **# eigenvectors predicted** | 6 | 6 | 5 | 5 |
| **Initial features** | Diffused dirac + Wavelet pos. | Diffused dirac + Wavelet pos. | Molecule features | Random |
| **MLP head type(s)** | Graph-level | Graph-level | Graph-level, per-node | Graph-level, per-node |
| **Graph-level MLP max # nodes** | 40 | 40 | 50 | 50 |
| **MLP head # layers** | 5 | 5 | 1 | 2 |
| **MLP head hidden dim** | 2400 | 2400 | N/A | 1600, 32 |
| **MLP head activation fn** | ReLU | ReLU | N/A | ReLU |
| **Loss function (and coefficient)** | 2*Eigenvector + 1*energy | 2*Eigenvector + 1*energy | 0.25 * Eigenvector + 0.05 * ortho | 0.25 * Eigenvector |
| **Other features/notes** |  | Removed graphs with less than six nodes during pre-training |  | Residual gating, virtual node |

## A.8 MODIFYING EXISTING PRE-TRAINING METHOD

### A.8.1 LEARNING RATE TUNING

We keep the majority of the settings from Hu et al. (2019) the same. For downstream fine-tuning, we tune over 7 learning rates for fair comparison according to Sun et al. (2022). We run each method and learning rate over 3 seeds, and select the learning rate based on mean validation accuracy over all learning rates.

### A.8.2 DOWNSTREAM DATASETS

We briefly list and cite the five downstream datasets here for reference. The five datasets are the datasets chosen in Sun et al. (2022), and are a subset of the eight primary downstream datasets evaluated in Hu et al. (2019).

- **BACE:** Qualitative binding results Subramanian et al. (2016)
- **BBBP:** Blood-brain barrier penetration Martins et al. (2012)
- **Tox21:** Toxicity data Mayr et al. (2016)
- **Toxcast:** Toxicology measurements Richard et al. (2016)
- **SIDER:** Database of adverse drug reactions (ADR) Kuhn et al. (2016)

## A.9 MODIFYING EXISTING GRAPH STRUCTURAL ENCODER

### A.9.1 LOSS FUNCTION COMPARISON

We claim that the sum of MAE and cosine similarity loss used in the structural encoder model GPSE (Cantürk et al., 2023) limits the model's ability to predict the eigenvectors of a graph.

To address sign ambiguity, the original GPSE model only learns the absolute value of each Laplacian eigenvector as its target. This improves performance over training without absolute value, but two shortcomings still arise. This (1) provides strictly less information than learning the true eigenvectors, and (2) does not capture basis ambiguity, making it ill-posed for eigenvectors with higher multiplicity (or eigenvectors with small eigengaps).

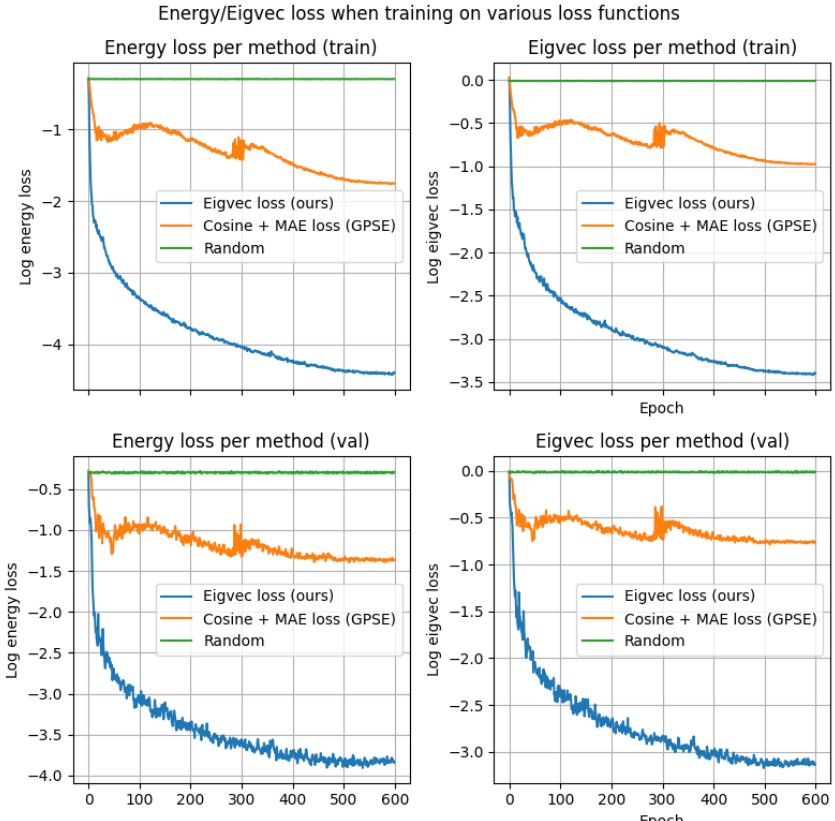

Figure 2: A comparison of energy loss and eigenvector loss, when the GPSE architecture is trained on (1) eigenvector loss (ours), (2) cosine similarity + MAE Cantürk et al. (2023), and (3) no loss (outputting random vectors). All losses displayed are in natural log scale for visual purposes.

We demonstrate this claim experimentally in Figure 2. We train the default GPSE settings and architecture using (1) eigenvector loss (ours), and the default cosine similarity + MAE loss (GPSE). We train for 600 epochs on 1% of the MolPCBA dataset and plot eigenvector and energy losses per-epoch. We also plot the loss when outputting random orthogonal vectors for comparison. While training on cosine similarity + MAE achieves better-than-random results, sign and basis ambiguity issues severely limit the model's ability to learn meaningful eigenvectors.