# OpenReview forum: "A Graph Laplacian Eigenvector-based Pre-training Method for Graph Neural Networks"
_ICLR.cc/2026/Conference — ICLR 2026 Conference Withdrawn Submission_

### Official Review · Reviewer_ip3E · 2025-10-26

**Soundness:** 2
**Presentation:** 1
**Contribution:** 3
**Rating:** 2
**Confidence:** 4

**Summary:**

This paper proposes LELM, a Laplacian eigenvector-based pre-training module for GNNs and graph foundation models. It leverages Laplacian eigenvectors and a global MLP head to capture long-range and global graph structure, addressing oversmoothing and expressivity limitations of standard GNNs. Experiments show that LELM improves downstream performance both as a standalone pre-training task and as an augmentation to existing pipelines.

**Strengths:**

1. LELM pre-trains GNNs to capture long-range and global graph structure, helping mitigate the oversmoothing problem and improving the expressivity of node representations.
2. LELM can be used either as a standalone pre-training task or as a plug-in to enhance existing graph pre-training pipelines, making it adaptable across different models and datasets.
3. The authors demonstrate the effectiveness of LELM across multiple downstream tasks, showing consistent performance improvements.

**Weaknesses:**

Although the paper addresses a meaningful problem, the writing and formatting appear underprepared. Each section has notable structural or formatting problems, raising concerns about whether the authors can adequately address these issues within the limited rebuttal period.

**Questions:**

1. The paper requires significant revision; for example, the introduction is scattered and does not clearly highlight the novelty, and the model framework figure should be redesigned for clarity.
2. Additional experiments are needed to demonstrate the effectiveness of the proposed method. Given the recent surge in GNN pre-training approaches—including graph prompting and graph foundation models—the current evaluation is weak and lacks comparisons with these strong baselines.
3. The paper should include a thorough analysis of time and computational complexity. Since the method involves graph spectra and Laplacian eigenvectors, scalability and efficiency are important concerns that are currently not addressed.

---

### Official Review · Reviewer_EUxQ · 2025-10-27

**Soundness:** 2
**Presentation:** 2
**Contribution:** 2
**Rating:** 4
**Confidence:** 4

**Summary:**

The paper introduces the Laplacian Eigenvector Learning Module (LELM), a self-supervised pre-training framework for Graph Neural Networks (GNNs). Instead of contrastive or feature-reconstruction objectives, LELM leverages graph Laplacian eigenvectors as pre-training targets to capture global and regional graph structures. The framework comprises three parts: (1) Node feature augmentation using diffusion-based embeddings; (2) Eigenvector prediction through a graph-level MLP head that aggregates all node embeddings; and (3) a joint loss combining energy loss and eigenvector loss. Experiments across molecular benchmarks show that LELM improves the downstream performance of GIN and GPS models and advances existing pre-training pipelines.

**Strengths:**

1. LELM defines eigenvector prediction as a self-supervised structural task, distinct from the traditional contrastive and reconstruction methods.

2. The results of LELM consistently outperform the base GNNs, e.g., GIN and GPS, demonstrating its effectiveness. Moreover, the ablation studies validate that reconstructing eigenvectors can help the model learn structural information.

**Weaknesses:**

1. The motivation of this paper is unclear. This paper states that GNNs struggle to capture global structures. The most popular solution is to use eigenvectors as the positional encoding to improve the expressive power of GNNs. However, it is unclear why we need to use the eigenvector reconstruction as a self-supervised target.

2. The novelty of this paper is limited. The base GNNs, i.e., GIN and GPS, are well-known GNNs. The auxiliary loss functions, i.e., energy loss and eigenvector loss, are proposed by the previous method. As a result, the paper does not appear to contain anything new, except for a pipeline.

3. Is the proposed method suitable for other tasks, such as node classification? Intuitively, introducing structural information will improve the performance of GNNs on the heterophilic node classification task. It would be better if the authors could provide more experimental results. This paper is currently incomplete.

4. Why does this paper use wavelet positional embeddings? Why not directly use the Laplacian eigenvectors as the positional embeddings? Does this lead to better performance?

**Questions:**

See weaknesses.

---

### Official Review · Reviewer_9rnk · 2025-10-28

**Soundness:** 2
**Presentation:** 2
**Contribution:** 2
**Rating:** 4
**Confidence:** 3

**Summary:**

The paper proposes LELM, a self-supervised, structure-aware pre-training module for GNNs that teaches a base GNN to predict the lowest-frequency Laplacian eigenvectors of each input graph. LELM is designed to be plug-and-play: it can serve as a standalone pre-training task or be added to existing graph pre-training pipelines and structural encoders.

**Strengths:**

easy to read

**Weaknesses:**

This paper seems not well developed. There is still a long way to go before publish.

**Questions:**

1. "Our use of the low-frequency graph Laplacian eigenvectors is motivated by their close relationship to structural and positional properties of graphs." The usefulness of low-frequency information is highly dependent on the tasks, i.e. the label distribution. In heterophilic graphs, it's not that effective.

2. Need a better description of the architecture, especially in section 4.2

3. Experimental results on node-level tasks, especially on the challenging heterophilic datasets listed in [1].




[1] Luan S, Hua C, Lu Q, Ma L, Wu L, Wang X, Xu M, Chang XW, Precup D, Ying R, Li SZ. The heterophilic graph learning handbook: Benchmarks, models, theoretical analysis, applications and challenges. arXiv preprint arXiv:2407.09618. 2024 Jul 12.

---

### Official Review · Reviewer_8iKV · 2025-10-31

**Soundness:** 2
**Presentation:** 1
**Contribution:** 1
**Rating:** 2
**Confidence:** 4

**Summary:**

In this work, the authors propose the Laplacian Eigenvector Learning Module (LELM), a novel pretraining module for graph neural networks (GNNs). Pre-training GNNs
to predict the low-frequency eigenvectors of the graph Laplacian matrix encourages the network to naturally learn large-scale structural patterns over each graph.

**Strengths:**

- Interesting approach to pre-train GNNs to predict the graph's Laplacian matrix eigenvectors

**Weaknesses:**

- Not very novel: existing work already uses Laplacian eigenvectors as node encodings (e.g., Dwivedi et al., ICLR 2021)
- Structure and writing of the paper can be significantly improved (no conclusion, single subsections 2.1 etc.)
- In general, the paper seems that requires a number of iterations (in related work, writing etc.)

**Questions:**

- line 46 missing reference?
- no reference of Figure 1 in the text
- do not add subsections if there is only one (e.g. 2.1)
- this reference appears twice in the References section: Vijay Prakash Dwivedi, Anh Tuan Luu, Thomas Laurent, Yoshua Bengio, and Xavier Bresson. Graph neural networks with learnable structural and positional representations. In International Conference on Learning Representations. Vijay Prakash Dwivedi, Anh Tuan Luu, Thomas Laurent, Yoshua Bengio, and Xavier Bresson.

---

### Note · Authors · 2025-11-14

I have read and agree with the venue's withdrawal policy on behalf of myself and my co-authors.